# GC-MS with Headspace Extraction for Non-Invasive Diagnostics of IBD Dynamics in a Model of DSS-Induced Colitis in Rats

**DOI:** 10.3390/ijms25063295

**Published:** 2024-03-14

**Authors:** Olga Yu. Shagaleeva, Daria A. Kashatnikova, Dmitry A. Kardonsky, Elena Yu. Danilova, Viktor A. Ivanov, Suleiman S. Evsiev, Eugene A. Zubkov, Olga V. Abramova, Yana A. Zorkina, Anna Y. Morozova, Dmitry N. Konanov, Artemiy S. Silantiev, Boris A. Efimov, Irina V. Kolesnikova, Julia A. Bespyatykh, Joanna Stimpson, Natalya B. Zakharzhevskaya

**Affiliations:** 1Laboratory of Molecular Pathophysiology, Lopukhin Federal Research and Clinical Center of Physical-Chemical Medicine of Federal Medical Biological Agency, Moscow 119435, Russia; olshagaleeva@gmail.com (O.Y.S.); daria_sv11@mail.ru (D.A.K.); freddy1178@yandex.ru (D.A.K.); ivanov@rcpcm.org (V.A.I.); efimov_ba@mail.ru (B.A.E.); irinakolesnikova@nextmail.ru (I.V.K.); juliabes@rcpcm.org (J.A.B.); 2Laboratory of Ecological Genetics, Vavilov Institute of General Genetics, Russian Academy of Sciences, Moscow 119991, Russia; 3Department of Analytic Chemistry, Faculty of Chemistry, Lomonosov Moscow State University, Moscow 119991, Russia; phenolyat@gmail.com; 4I.M. Sechenov First Moscow State Medical University, Moscow 119048, Russia; 5Department of Basic and Applied Neurobiology, V. P. Serbsky National Medical Research Center for Psychiatry and Narcology, Moscow 119034, Russiazorkina.ya@serbsky.ru (Y.A.Z.);; 6Laboratory of Mathematical Biology and Bioinformatics of Scientific Research Institute for Systems Biology and Medicine, Moscow 117246, Russia; konanovdmitriy@gmail.com; 7School of Health Sciences, Faculty of Biology, Medicine and Health, University of Manchester, Manchester M13 9PL, UK

**Keywords:** volatile organic compounds, HS-GC-MS, IBD, DSS-induced colitis

## Abstract

Inflammatory bowel diseases are extremely common throughout the world. However, in most cases, it is asymptomatic at the initial stage. Therefore, it is important to develop non-invasive diagnostic methods that allow identification of the IBD risks in a timely manner. It is well known that gastrointestinal microbiota secrete volatile compounds (VOCs) and their composition may change in IBD. We propose a non-invasive method to identify the dynamics of IBD development in the acute and remission stage at the level of VOCs in model of dextran sulfate sodium (DSS) with chemically induced colitis measured by headspace GC/MS (HS GC/MS). Methods: VOCs profile was identified using a headspace GC/MS (HS GC/MS). GC/MS data were processed using MetaboAnalyst 5.0 and GraphPad Prism 8.0.1 software. The disease activity index (DAI) and histological method were used to assess intestinal inflammation. The peak of intestinal inflammation activity was reached on day 7, according to the disease activity index. Histological examination data showed changes in the intestine due to different stages of inflammation. As the acute inflammation stage was reached, the metabolomic profile also underwent changes, especially at the short-fatty acids level. A higher relative amounts of acetic acid (*p* value < 0.025) and lower relative amounts of propanoic acid (*p* value < 0.0005), butanoic acid (*p* value < 0.005) and phenol 4-methyl- (*p* value = 0.053) were observed in DSS7 group on day 7 compared to the control group. In remission stage, disease activity indexes decreased, and the histological picture also improved. But metabolome changes continued despite the withdrawal of the DSS examination. A lower relative amounts of propanoic acid (*p* value < 0.025), butanoic acid (*p* value < 0.0005), pentanoic acid (*p* value < 0.0005), and a significant de-crease of hexanoic acid (*p* value < 0.0005) relative amounts were observed in the DSS14 group compared to the control group on day 14. A model of DSS-induced colitis in rats was successfully implemented for metabolomic assessment of different stages of inflammation. We demonstrated that the ratios of volatile compounds change in response to DSS before the appearance of standard signs of inflammation, determined by DAI and histological examination. Changes in the volatile metabolome persisted even after visual intestine repair and it confirms the high sensitivity of the microbiota to the damaging effects of DSS. The use of HS GC/MS may be an important addition to existing methods for assessing inflammation at early stages.

## 1. Introduction

Inflammatory bowel diseases (IBD) are the main problem of modern gastroenterology, represented by two types of diseases—ulcerative colitis (UC) and Crohn’s disease (CD) [1]. To date, there are no reliable methods for the early diagnosis of IBD. In addition, there are considerable difficulties to relieve the most severe symptoms of the disease, such as bloody diarrhea, which contributes to exhaustion and death. In most cases, there are no significant symptoms in the initial stages of IBD, so patients do not seek help in a timely manner. On the other hand, the diagnostic methods that often used when the symptoms of the disease are pronounced are associated with various procedural complications, such as functional impairment, pain reactions, respiratory distress, and bacterial infections, which can occur both during and after the invasive procedure [2]. Considering these reasons, it is necessary to search for alternative routes of IBD diagnosis and treatment. Molecular methods are the most promising for disease verification since they do not involve any risk for the patient and can be carried out in a relatively short time period.

Metabolomics is one of the tools for assessing a set of diagnostic markers for IBD verification. It is well-known that microorganisms secrete a significant number of compounds including volatile organic compounds (VOC), which can be found in stool [3]. Short-chain fatty acids are the main microbial metabolites [4]. In particular, the acetic, propionic, and butyric acids produced by the microbiota make up 90–95% of the VOC in the colon (ratio 3:1:1) [5]. In addition, medium and long-chain fatty acids, amino acids derivatives, etc., are also included as volatile metabolites. For most volatile metabolites, specific biological effects have not yet been observed, but according to the obtained data, the ratio of the components of the volatile metabolome in the samples is associated with the development stage of intestine inflammation [6,7]. Analysis of faecal volatile organic metabolites can provide an understanding of gut metabolomic changes in IBD and has the potential to provide a non-invasive means of diagnosing IBD, contributing to differentiate between UC and CD. In one of the research it has been shown that Heptanal, 1-octen-3-ol, 2-piperidinone and 6-methyl-2-heptanone were up-regulated in the active CD group while methanethiol, 3-methyl-phenol, short-chain fatty acids and ester derivatives were found to be less abundant [8]. Walton et al. found the increased concentrations of indoles, alcohols and esters in patients with active CD. After proper treatment of these patients, concentrations were closer to those of healthy individuals. So this study outcomes imply that faecal VOC analysis holds potential for identifying biomarkers for IBD detection and for monitoring disease activity [9]. Gas chromatography combined with mass spectrometry is usually used for VOC annotation [10,11]. The vapor-phase extraction method realized by headspace GC/MS (HS GC/MS) makes it possible to compare various biological samples, including feces, since the ratios of components in an equilibrium vapor do not depend on the amount of water contained in the samples [12]. As previously described, it is difficult to annotate compounds at the early stage of inflammation in real patients; therefore, animal experimental models with chemically induced colitis can prove the effectiveness of the technique at different stages of pathology.

Chemically induced rodent models are the most precise for IBD researches. A number of chemicals are used to model an inflamed bowel including 2.4.6-trinitrobenzene sulfonic acid (TNBS), oxazolone and sodium dextran sulfate (DSS). The TNBS model causes transmural colitis, which is due to the TH1-mediated immune response. Colitis induced by oxazolone, suspended in ethanol, unlike the TNBS-induced colitis, contributes to the TH-2-type cytokine response, with surface lesions of the mucosa of the distal colon. Nonetheless, the DSS model is the most frequently used chemical in modeling colitis in rodents. Unlike two other models, it is well suited for studying congenital immunity and healing processes, due to good reproduction, ease of modification of doses, and treatment/induction cycles [1,13,14,15,16,17]. The oral administration of DSS also simplifies the course of the experiment. However, DSS causes damage to the epithelium of the colon in all its departments. The recent results also showed that sulfate groups of DSS molecules destabilize the layers of mucus and increase the permeability of the colon, destroying dense contacts between epithelial cells, reducing mucin levels, and simultaneously changing the resident microbiota [18,19]. The intestinal microbiota and metabolites are involved in the creation of a biological barrier, and perform the functions of biological antagonism, such as protection against infection, participation in immune system maturation; regulation of intestinal mucus production; regulation of intestinal epithelium nutrition and metabolism [4]. Microbiota and their metabolites are major factors regulating the composition and function of mucus, in turn influencing the function and structure of intestinal mucus barrier under colitis. It has been shown that pseudo-aseptic mice, treated with antibiotics and were more severely ill than mice with intact microbiota and their intestinal mucus loss was more pronounced in DSS-induced colitis (*p* < 0.05), suggesting that different microbiota and metabolites could cause the different degrees of colitis. Subsequently, authors observed that *Klebsiella* and *Bacteroides*, were widely associated with colitis, while *Candidatus Stoquefichus*, *Anaerobiospirillum*, *Muribaculum*, and *Negativibacillus* were involved in protection against colitis. Furthermore, differential metabolites of the microbiota were mainly enriched in the synthesis-related pathways of key structural sequences of mucin. In combination with the mucin-related staining and immunofluorescence results indicated that the differential microbiota and their metabolites potentially regulate the composition and function of mucus under colitis [20]. Taking into account the classic ways of experimental colitis confirmation, such as histological assay and the disease activity index (DAI), analysis of microflora metabolites can assess the degree of damage and subsequent restoration of intestinal tissue [1,19,21]. Discovering alternative methods of studying the degree of regeneration of the intestinal mucosa after exposure to the chemical agent is important in diagnosing the severity of damage to the intestinal wall and assessing the effectiveness of therapeutic agents, such as probiotics or metabolites [22].

Thus, in this study, we pursued the goal of creating DSS-induced colitis in rats and subsequent analysis of intestine inflammation and regeneration stages by using HS GC/MS method combined with histological examination of the intestinal tissue and evaluating the index of activity of the disease.

## 2. Results

### 2.1. Histopathological Changes in Acute Phase and in Remission

The total score of DAI consisted of body weight decrease, stool consistency, and rectal bleeding [20,23,24]. There was no weight decrease in the control group of the animals or changes in stool throughout the experiment. As shown in Figure 1A, the DAI score was kept close to 0 in the control group during all experimental procedures. There was a noticeable weight decrease in animals on the 3rd experimental day in DSS7 and DSS14 groups. Further, by day 7, weight continued to decrease, the consistency of the stool became soft, streaked with visible blood. The DAI of the DSS-treated groups (DSS7 and DSS14) was significantly higher compared to the control group and increased on day 3. The maximum scores counted for the DSS-treated groups (DSS7 and DSS14) were obtained on day 7 when the acute phase of inflammation was reached.

According to histopathological assay (Figure 1B,C), the colons of the untreated rats had intact mucosa, whereas DSS-treated colons demonstrated inflammatory cell infiltration, thinning of the mucosa and submucosa, and epithelium erosion (Appendix A). The histopathological scoring system was used to detect epithelial damage as well as inflammatory cell infiltration (Figure 1B).

In the DSS7 group, almost no goblet cells were detected, only single crypts remained, marked inflammatory infiltration, and the formation of single crypt abscesses were observed.

The capability of the intestines to recover was noted on day 14, when there was a normalization of stool consistency and restoration of animal’s weight in the DSS 14 group. There was no visible blood in the stool, and higher activity of the animals was observed. As expected, the DAI values decreased in the DSS14 group after stopping the DSS administration on 8–14 days (Figure 1A). A definite improvement in the histological picture was observed in the DSS14 group, in which the number of crypts and goblet cells increased and the mucosal structure was more differentiated, however minimal inflammatory infiltration was preserved (Figure 1C).

As expected, the DAI values decreased in the DSS14 group after stopping the DSS administration on 8–14 days (Figure 1A). The capability of the intestine to recover on day 14 was noted, when normalization of stool consistency and regain weight of animals were observed. There was no visible blood in the stool and higher activity of the animals was observed. A definite improvement in the histological picture was observed in the DSS14 group, in which the number of crypts and goblet cells increased and the mucosal structure was more differentiated; however, minimal inflammatory infiltration was preserved (Figure 1C).

### 2.2. Metabolomic Data from Acute Phase to Remission Stage

Metabolomic data of stool volatile compounds were normalized (Appendix A). At least 60% of stable detected compounds were used for metabolomic profile comparisons. The 16 most commonly detected compounds that passed through the selection criteria are presented in the Table 1. Among the identified compounds SCFA, medium and long-chain fatty acids were annotated. Differences in the metabolomic profiles of the control group and the DSS7 and DSS14 groups were shown using PCA (Figure 2A). Changes in the metabolomic profile can be observed by the third day after DSS administration (Figure 2B). However, the main colitis symptoms, such as diarrhea and blood in the stool were not yet detected within this time period (Figure 1A).

The acute phase of colitis peaked on the seventh day after DSS administration and was characterized by stool changes. Animals already had blood in stool and diarrhea before day 7. The animals were observed for a recovery period of 7 days following the administration of DSS, which is denoted as day 11 and 14 in the experiment and graphs. The results of metabolite content analysis in each animal sample relative to the day of selection are represented by each point on the graphs (Figure 3A,B). The detected compounds that strictly fulfilled the criteria described in the previous paragraph are listed in Table 1. The metabolite profile of the samples was scored based on the relative content of the 10 most frequently identified compounds.

Despite the improvement shown by the histological picture after stopping the DSS administration, there was a trend towards the progression of VOC profile changes.

Metabolomic changes continued to increase by day 14. As it can be seen in Figure 3A,C there were significant changes in the metabolomic profile for such compounds as acetic acid, butanoic acid, pentanoic acid, propanoic acid, propanoic acid 2-methyl-, and hexanoic acid, registered in the DSS14 group. Changes in metabolites relative amountsm such as benzenepropanoic acid and indolem were noted on day 11. But, at day 14, significant changes in the mentioned metabolites relative amounts could not be assessed. (Table 1).

Therefore, according to the VOC metabolome data, it can be assumed that the microbiota structure continued to change up to the 14th day in contrast to the obvious symptoms of colitis, which were subsiding (Figure 1A,B and Figure 3).

## 3. Discussion

Bacterial population shifts are usually observed in IBD and contribute to changes in metabolite levels [25]. Most of the metabolites released by bacteria are volatile compounds and changes of their relative amounts are the main sign of restructuration in microbiota. In our study, we hypothesized that even mild colonic inflammation can influence the structure of the microbiota and, as a consequence, contribute to changes in the ratio of released metabolites. The DSS-induced colitis model is ideal for dynamic monitoring of intestine inflammation development and for recording of metabolomic changes in stool. The basic tools for assessing inflammation in such models are histological methods. A method for assessing clinical manifestations is the disease progression index (DAI), which includes such criteria as animal weight, stool consistency and blood presence, usually used. Nevertheless, mentioned histological methods may not be sufficient for a full understanding of pathology changes in the created experimental colitis model, especially for early detection of inflammation. We believe that microbiota functional activity can contribute to a more detailed description of the inflammation and recovery processes in animal models. The analysis of volatile components secreted by microbiota can act as an additional criterion for assessing the specific stages of colitis.

In this study, the DSS-induced colitis model was selected because of its convenience and availability. The optimal concentration of 4.5% DSS was selected, which allowed the smooth formation of the inflammatory process within 7 days without high mortality caused by intestinal necrosis. Furthermore, after the discontinuing of DSS, the animals were observed for an additional 7 days in order to evaluate the spontaneous recovery of intestinal tissues and compare the levels of measurable metabolites in stool samples during the inflammation and recovery stages. The GC/MS method with headspace extraction was employed in this study, which made it possible to carry out analysis regardless of the consistency of the stool samples.

It was found that the metabolomic profiles of the experimental groups differed from those of the control group. The principal component analysis (PCA) gave the first informative look at the structure of the dataset and the relationships between groups. We recorded metabolomic changes already on the third day of the experiment, while the DAI did not correspond to the values recorded during the colitis formation. During the formation of the colitis model on days 1–7, significant changes were observed only in the most represented metabolites, such as acetic and butanoic acid. Over the next 8–14 days, after the discontinuation of DSS administration, there was a decrease in DAI values, and intestinal recovery was observed, according to histological data. However, the VOCs metabolomics changes continued to progress in the DSS14 group. A relative reduction in metabolites was observed not only for acetic acid, butanoic acid, but also for pentanoic acid, propanoic acid, propanoic acid 2-methyl-, and hexanoic acid. The decrease in the metabolites relative amounts was probably associated with ongoing intestinal bacterial community changes.

The main producers of SCFAs are Bacteroidetes (Gram-negative) and Firmicutes (Gram-positive), which are the most common bacterial types of the microbiota. Bacteroidetes mainly produce acetate and propionate, while Firmicutes produce butyrate [26]. Bacteroides species secreted glycoside hydrolases and polysaccharide lyases and act as primary degraders of complex polysaccharides, releasing simpler oligosaccharides and metabolites used by secondary fermenters, such as Clostridium and Lactobacillus. These cross-feeding mechanisms result in the formation of acetate, propionate, butyrate, and (in lower proportion) valerate [27,28]. It is also known that short-chain fatty acids and branched-chain fatty acids are also produced by *Acidaminococcus* spp., *Acidaminobacter* spp., *Campylobacter* spp., *Eubacterium* spp., *Fusobacterium* spp., and *Peptostreptococcus* spp. [29,30]. Although the majority of the SCFAs are derived from dietary fiber, another source of microbial accessible carbohydrates (MACs) is the colonic mucus layer. Mucus is secreted by goblet cells in both the small and large intestine and comprises large proteins (>5000 amino acids). In the DSS-induced colitis model, it was observed that there was a lack of dietary fiber, as well as destabilization of mucin, due to the action of DSS and destruction of goblet cells. Thus, a change in the relative amounts of metabolites may be a consequence of both changes in the structure of the microbiota and damage to the mucin layer. However, changes in the levels of metabolites were detected already on the third day of the experiment, as well as on the 14th day of the experiment, when the DAI was close to the control values. Thus, it can be assumed that the minimal inflammation that formed under the DSS exposure at the early stages of developing colitis can be recorded by detection of volatile compounds relative amounts changes. DSS possibly affects the structure of the microbiota from the first days of exposure and its impact is reflected in the levels of metabolites. In addition, changes in the microbiota composition can continue even after mucosa recovery, which is reflected in the quantitative ratio of metabolites. Therefore, it is possible to assess the likelihood of complete intestine restoration only after normalization of the ratio of volatile metabolites.

DSS-induced colitis model in animal is only a one-sided view of the real IBD. Among the main reasons for the formation of colitis there are also genetic characteristics and autoimmune components, which are not taken into account within this model. However, when monitoring the microbiota composition of IBD patients we can observe the quantitative alterations in fecal metabolites. Accordingly, it can be assumed that the microbiota, as a highly sensitive indicator, responds to any triggering factors that leads to the epithelium damage. In this regard, HS-GC/MS method is also justified in the clinic during routine medical examination, since it can be assumed that even before the appearance of clinical symptoms, the spectrum of metabolites may undergo alterations. Taking into account the data obtained, it is planned to use HS-GC/MS method for the clinical examination of patients with suspected inflammatory bowel diseases. Already today, headspace extraction method is actively used to analyze the volatile components of the IBD patient’s metabolome to analyze the effectiveness of the treatment in combination with the regular blood and fecal tests.

## 4. Materials and Methods

### 4.1. Animals

We used female Wistar rats from the Nursery for Laboratory Animals (Pushchino, Russia) in the experiment that were 2 months old (200–230 g). The animals were randomly divided into three groups: the control group (*n* = 10), the dextran sulfate sodium (DSS7) group (*n* = 5), and the dextran sulfate sodium (DSS14) group (*n* = 5). Each rat in the DSS7 group was orally administered 4.5% DSS for 7 days, and afterwards they were all euthanized by chloroform. Each rat in the DSS14 group was orally administered 4% DSS for 7 days, pure water for the next 7 days, and then were all euthanized by chloroform. The control group rats were administered the same volume of normal saline for 14 days before euthanasia by chloroform. Body weights, stool, consistency, and the presence of occult blood were measured every 3 days.

### 4.2. Ethics Statement

All experimental procedures were set up and maintained in accordance with Directive 2010/63/EU of 22 September 2010 and approved by the local ethical committee of V.P. Serbsky National Medical Research Center for Psychiatry and Narcology.

### 4.3. Reagents

Dextran sulfate sodium salt, Mr ~40,000, Alfa Aesar. N, O-bis(trimethylsilyl)trifluoroacetamide (with 1% trimethylchlorosilane; *v*/*v*; lot no. B-023), and heptadecanoic acid (as an internal standard, IS, purity ≥ 98%; lot no. H3500) were purchased from Sigma-Aldrich (Saint Louis, MI, USA). O-methylhydroxylamine hydrochloride (ity: 98.0%; lot no. 542171) was purchased from J&K Scientific Ltd. (Beijing, China). Pyridine (lot no. C10486013) was obtained from Macklin Biochemical (Shanghai, China). Chromatographic-grade methanol was purchased from Thermo Fisher Scientific (Waltham, MA, USA). Pure water was obtained from the Wahaha Company (Hangzhou, China).

### 4.4. Clinical Disease Score

The disease activity index (DAI) of the rats was estimated by the score of body weight loss (no weight loss: 0; 5–10% weight loss: 1; 11–15% weight loss: 2; 16–20% weight loss: 3; 20% weight loss: 4); stool consistency (formed: 0; watery stool: 2); and the degree of stool occult blood (normal color stool: 0; reddish color stool: 2). The DAI was tested at the last day of the experiment to assess the overall disease severity.

### 4.5. Sample Preparation

For HS-GC/MS analysis, rats stool samples (all available sample volume) were collected every 3 days in the control, DSS7, and DSS14 groups.

### 4.6. HS-GC/MS

In total, 50–100 mg stool samples plus 500 µL water samples were placed into 10 mL screw-cap vials for a Shimadzu (Kyoto, Japan) HS-20 headspace extractor. A total of 0.2 g of a mixture of salts (ammonium sulfate and potassium dihydrogen phosphate in a ratio of 4:1) was added to increase the ionic strength of the solution. The headspace extractor settings used were oven temperature 80 °C, sample line temperature 220 °C, transfer line temperature 220 °C, equilibrating time 15 min, pressurizing time 2 min, load time 0.5 min, injection time 1 min, and needle flush time 7 min. The vials were sealed and analyzed on a Shimadzu QP2010 Ultra GC/MS with a Shimadzu HS-20 headspace extractor, a VF-WAXMS column with a length of 30 m, a diameter of 0.25 mm, and a phase thickness of 0.25 microns. The initial column temperature was 80 °C, heating rate 20 °C/min to 240 °C, exposure 20 min. The carrier gas was helium 99.9999, the injection modewas splitless, and the flow rate was 1 mL/min. The ion source temperature was 230 °C. The interface temperature was 240 °C. The total ionic current (TIC) monitoring mode was used. To analyze the obtained mass spectra, the NIST 2014 mass spectra library with automated mass spectral deconvolution and identification system (AMDIS version 2.72) was used.

### 4.7. Histology

The samples of colons were cut lengthwise, washed with PBS, and rolled up as a “Swiss roll” [31]. The obtained material was fixed in 10% buffered formalin, pH 7.0–7.8, for 48 h, and then placed in labeled histologic cassettes with a liner to prevent unrolling. Histologic wiring was performed with isopropyl alcohol (Chimmed, Moscow, Russia) and two changes of o-xylene (Chimmed, Russia). The processed samples were embedded in paraffin wax (BioVitrum, Moscow, Russia). Slices with a thickness of 3.5 μm were prepared on a rotary microtome Thermo HM 340E (Thermo Scientific, Beijing, China, manufacturer) as that slice plane was perpendicular to the axis of the folding specimen. After drying, the sections were stained with hematoxylin-eosin (BioVitrum, Russia). The specimens were visualized by using light microscopy (Zeiss Primo Star, Shanghai, China). A ball-based scoring system was used to assess the damage severity as previously described [32].

### 4.8. Data Processing

The HS-GC/MS data were processed as follows. Peak areas computed by AMDIS for the selected compounds were converted to relative abundances. The volatile compound percentages were estimated by summing up the percentage of confidently identified compounds for each sample in the AMDIS database. The resulting values were then recalculated as a percentage of the total reliably identified compounds. These conversions were required to avoid content estimation errors due to unreliable matrix signals caused by noise.

### 4.9. Statistical Analysis

The GC-MS data were processed using MetaboAnalyst 5.0 (http://www.metaboanalyst.ca accessed on 20 February 2023) and GraphPad Prism 8.0.1 software. The values obtained for each animal were considered paired and consistent, with confirmation by the ROUT outlier test (ROUT (Q = 1%)). Since it was not possible to assess the normality of distribution for all sample groups, it was assumed that the raw data did not conform to a normal distribution. The nonparametric Mann–Whitney test was used to conduct the primary comparison between groups. After natural log transformation, normalized data were analyzed using a standard *t*-test and ANOVA. Statistical significance was determined by a two-sided *p*-value of less than 0.05. To reduce the number of dimensions and explore the data further, an unsupervised principal component analysis (PCA) was performed. Prior to the PCA, standard procedures were executed. A Chi-squared test was used to identify a method to complete missing values. The missing at random (MAR) criteria were identified in the data. The BPCA method was determined to be the most suitable. Subsequently, normalization and scaling procedures were executed.

## 5. Conclusions

As a result of our study, a model of DSS-induced colitis in rats was successfully implemented for metabolomic assessment of different stages of inflammation. We demonstrated that the ratios of volatile compounds changed in response to DSS before the appearance of standard signs of inflammation, determined by DAI and histological examination. Changes in the volatile metabolome persisted even after visual intestine repair and it confirmed the high sensitivity of the microbiota to the damaging effects of DSS. The use of HS GC/MS may be an important addition to existing methods for assessing inflammation at early stages.

## Figures and Tables

**Figure 1 ijms-25-03295-f001:**
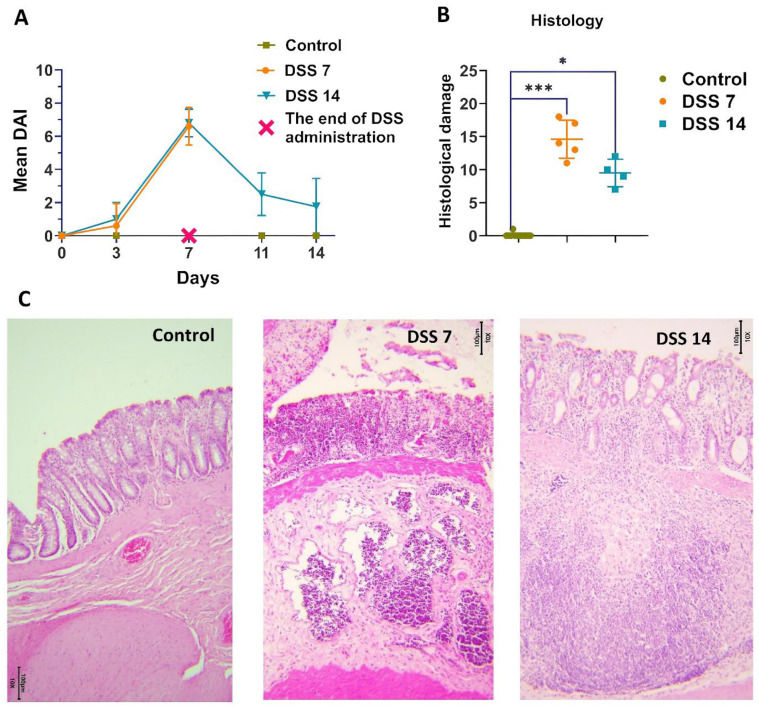
Development of DSS-induced colitis in different phases of inflammation. (**A**) Dynamics of the disease activity index (DAI) in each group. (**B**) Histopathological evaluation of the colon of each group of rats. Data are presented as individual means or mean ± SD for each group * (*p* < 0.025), and *** (*p* < 0.0005). (**C**) Colon tissue stained with H&E (10×).

**Figure 2 ijms-25-03295-f002:**
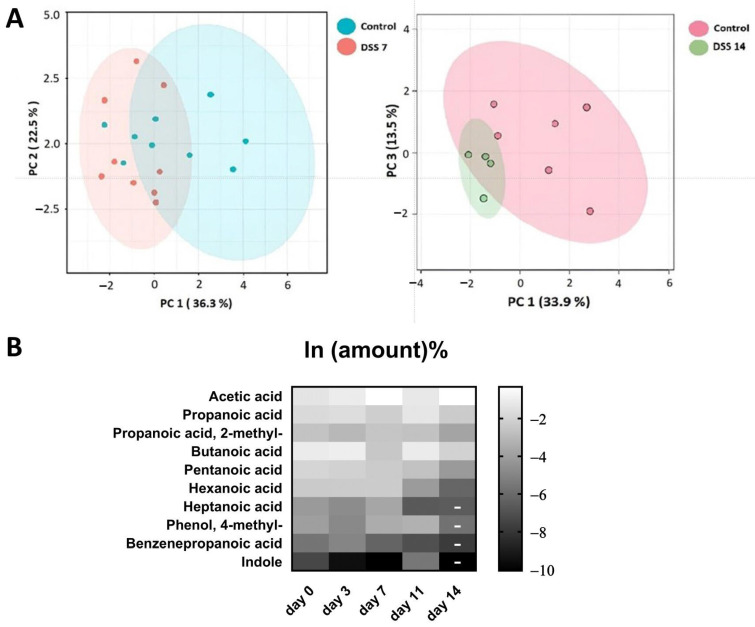
(**A**). Comparison of VOC metabolomic profiling data using principal component analysis (PCA) between experimental and control groups. (**B**) Changes in the metabolomic profile during the experiment. Due to their wide range, the data were log-transformed. On day 14, the amounts of metabolites identified in the small number of samples are indicated by dashes and were not analyzed for significance.

**Figure 3 ijms-25-03295-f003:**
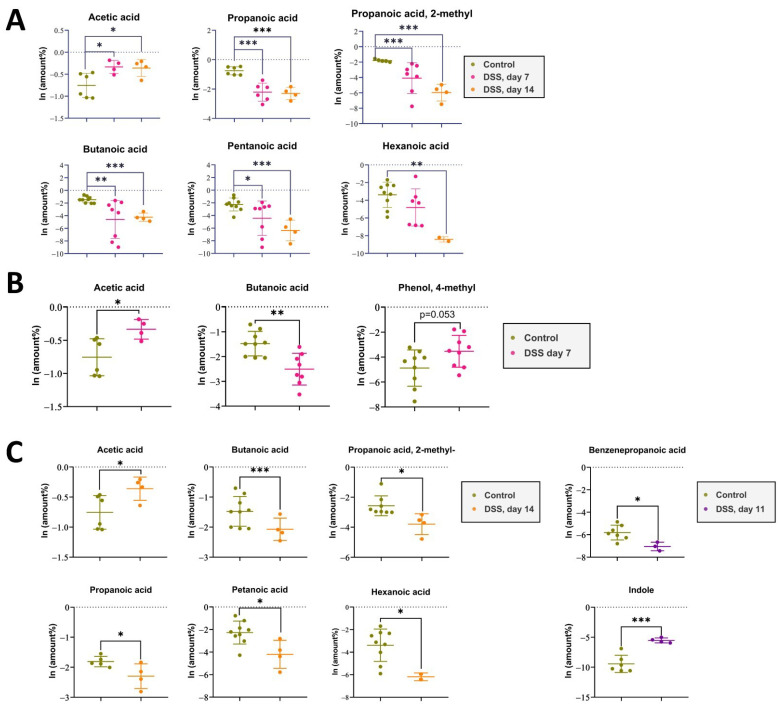
(**A**). The impact of DSS administration on short-chain fatty acid levels in rat feces was assessed at day 7 of DSS intake and day 14 after 7 days of recovery, compared to the control group. Data represent means ± S.D. Statistically significant differences are indicated by * (*p* < 0.025), ** (*p* < 0.005), and *** (*p* < 0.0005) evaluated by ANOVA. (**B**). The impact of DSS administration on short-chain fatty acid levels in rat feces was assessed at day 7 of DSS intake compared to the control group. Data represent means ± S.D. Statistically significant differences are indicated by * (*p* < 0.025), ** (*p* < 0.005) evaluated by *t*-test. (**C**). The impact of DSS administration on short-chain fatty acid levels in rat feces was assessed at day 14 after 7 days of recovery, compared to the control group. Data represent means ± S.D. Statistically significant differences are indicated by * (*p* < 0.025) and *** (*p* < 0.0005) evaluated by *t*-test.

**Table 1 ijms-25-03295-t001:** List of the 10 most abundant compounds in rat stool samples selected for analysis. Data represent means ± S.D. Statistically significant differences are evaluated by non-parametric Mann–Whitney test.

Metabolite	Percent of Samples in Which the Marker Is Detected Out of a Total of *n* = 44	Control Group, Amount%	Group DSS, Day 0, Amount%	Group DSS, Day 3, Amount%	Group DSS, Day 7, Amount%	Group DSS, Day 11, Amount%	Group DSS, Day 14, Amount%
Acetic acid	65	26.42 ± 7.24	34.83 ± 7.97	72.21 ± 10.5	34.71 ± 20.66	70.72 ± 12.6	48.55 ± 13.3
Propanoic acid	70	18.26 ± 7.48	19.99 ± 5.14	12.73 ± 7.36	30.4 ± 16.08	10.69 ± 4.19	16.58 ± 3
Propanoic acid, 2-methyl-	91	7.52 ± 2.36	5.2 ± 2.61	10.93 ± 9.18	7.61 ± 4.31	2.61 ± 1.39	5.63 ± 0.98
Butanoic acid	98	33.96 ± 6.73	37.3 ± 8.6	9.66 ± 5.81	33.01 ± 4.61	13.3 ± 5.31	25.51 ± 12.93
Pentanoic acid	100	14.47 ± 3.68	12.65 ± 3.85	11.39 ± 7.4	8.00 ± 5.26	2.44 ± 2.51	15.30 ± 14.02
Hexanoic acid	77	10.23 ± 4.63	10.36 ± 3.07	21.71 ± 22.61	2.20 ± 2.22	0.21 ± 0.07	6.35 ± 5.92
Heptanoic acid	73	1.81 ± 1.19	1.08 ± 0.63	6.44 ± 7.53	0.31 ± 0.40	0.14 ± 0.00	1.13 ± 0.75
Phenol 4-methyl-	93	2.20 ± 1.30	1.21 ± 1.81	5.49 ± 6.11	5.24 ± 5.37	0.32 ± 0	1.44 ± 1.33
Benzenepropanoic acid	84	0.39 ± 0.17	0.97 ± 1.12	0.27 ± 0.27	0.09 ± 0.03	0.04 ± 0.00	0.36 ± 0.23
Indole	75	0.11 ± 0.12	0.10 ± 0.1	0.01 ± 0.02	0.42 ± 0.18	0.004 ± 0.000	0.01 ± 0.01

## Data Availability

Data are contained within the article and Appendix A.

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
