# Peer review of "GC-MS with Headspace Extraction for Non-Invasive Diagnostics of IBD Dynamics in a Model of DSS-Induced Colitis in Rats"

_ijms, 2024, doi:10.3390/ijms25063295_

Round 1
Reviewer 1 Report
Comments and Suggestions for Authors
The manuscript (ijms-2887255-peer-review-v1) entitled ‘GC-MS with Headspace Extraction for non-invasive Diagnostics of IBD dynamics in a Model of DSS-induced Colitis in 3 Rats’ has a few major issues (listed below).
Major comment: The major concern is in Sections 2.6 and its results presented in Table 1. In the HS-GC/MS system, the volatiles were extracted, analysed through a GC/MS and the metabolites were identified through mass spectral matching with databases. The major identified metabolites in control and test samples are presented in Table 1. The data in Supplementary Table 2 is also derived in the same pattern. In gas chromatographic experiments, identification of metabolites by database matching is only the basic method. This (identification) is to be further supported by secondary and tertiary confirmations such as LRI determination and co-gas chromatography (with standards). Table 1 is listing the 10 major metabolites; these compounds could have been further confirmed by co-GC under similar GC experimental conditions. Without these data, the identifications listed in Table 1 are not reliable. These data (volatile metabolites in Table 1) are very crucial in the whole theme of the manuscript. Therefore, I suggest the authors to explain this and provide secondary/tertiary conformation on the identification of the major compounds listed in Table 1.
Minor comments: Please do not start sentences with numbers (for example Section 2.6). Table 1: present the percentages in decimals (example, 26.42 ± 7.24). The percentage decimal accuracy in a GC-MS determination is to be restricted to one (example, 26.4 ± 7.2).
I suggest a major revision of the manuscript based on the above major comments.
Comments on the Quality of English LanguageModerate editing of English language required
Author Response
Dear colleague,
We are very thankful for this question since its excited us for a long time. We used AMDIS software for identified compounds detected by HS GC/MS method. AMDIS software identify the found compounds based on their purified spectra and retention indices and characterizes the quality of identification. We conducted a detailed study of the applicability of the headspace extraction method for the analysis of complex mixtures, before research we present you [1]. In our previous work, we analyzed synthetic mixtures using three methods: NMR, GC-FID and HS GC/MS. We became confident in the reliability of the compounds identification by obtaining identical data for the analyzed mixtures of components in different ratios using NMR, GC-FID and HS GC/MS. Moreover, we have developed an approach that allows us to convert quantitative data from the vapor to liquid phase. Next, we moved on to the analysis of fecal samples and obtained convincing data on the convergence of NMR, GC-FID and HS GC/MS methods in terms of identifying individual compounds in complex mixtures [1,2]. We also relied on protocols for analyzing of fecal samples presented in the literature when developing a headspace analysis technique [3,4]. Using HS GC/MS and NMR, we also annotated compounds of bacterial media, confirming the convergence of identification of VOCs components by HS GC/MS and NMR [5]. We also provided the study of the Bacteroides secretome confirming the applicability of the HS GC/MS method for the analysis and identification of components of complex mixtures [6]. Like in our research, in literature there are available improved data on the use of the HS GC/MS method for fecal analysis and VOCs identification [7].
- Dmitry N Konanov , Natalya B Zakharzhevskaya , Dmitry A Kardonsky , Elena S Zhgun , Yuri V Kislun , Artemy S Silantyev , Olga Yu Shagaleeva , Danil V Krivonos , Alexandra N Troshenkova , Vadim M Govorun , Elena N Ilina UniqPy: A tool for estimation of short-chain fatty acids composition by gas-chromatography/mass-spectrometry with headspace extraction J Pharm Biomed Anal. 2022 Apr 1:212:114681.
- Qin-Yuan Hu, De-Rong Huang, Shao-Kun Wang, A new method for predicting liquid-liquid equilibria from headspace analysis determination over homogeneous liquid mixtures using the uniquac model, Fluid Phase Equilib. 84 (1993) 267–280
- Naama Karu, Lu Deng, Mordechai Slae, An. Chi Guo, Tanvir Sajed, Hien Huynh, Eytan Wine, David S. Wishart, A review on human fecal metabolomics: Methods, applications and the human fecal metabolome database, Anal. Chim. Acta. (2018) 1–24
- Sofia El. Manouni El Hassani, Ruud J. Soers, Daniel J.C. Berkhout, Hendrik J. Niemarkt, Hans Weda, Tamara Nijsen, Marc A. Benninga, Nanne K.H. de Boer, Tim G.J. de Meij, Hugo H. Knobel, Optimized sample preparation for fecal volatile organic compound analysis by gas chromatography-mass spectrometry, Metabolomics 16 (10) (2020 10) 112,
- Chaplin AV, Shcherbakova VA, Pikina AP, Sokolova SR, Korzhanova M, Belova VA, Korostin DO, Rebrikov DV, Kardonsky DA, Urban AS, Zakharzhevskaya NB, Suzina NE, Podoprigora IV, Das MS, Kholopova DO, Efimov BA. Diplocloster agilis gen. nov., sp. nov. and Diplocloster modestus sp. nov., two novel anaerobic fermentative members of Lachnospiraceae isolated from human faeces. Int J Syst Evol Microbiol. 2022 Feb;72(2). doi: 10.1099/ijsem.0.005222.
- Shagaleeva OY, Kashatnikova DA, Kardonsky DA, Konanov DN, Efimov BA, Bagrov DV, Evtushenko EG, Chaplin AV, Silantiev AS, Filatova JV, Kolesnikova IV, Vanyushkina AA, Stimpson J, Zakharzhevskaya NB. Investigating volatile compounds in the Bacteroides secretome. Front Microbiol. 2023 May 3; 14:1164877. doi: 10.3389/fmicb.2023.1164877. PMID: 37206326; PMCID: PMC10189065.
- C. Zhang, P. Tang, H. Xu, et al., Analysis of short-chain fatty acids in fecal samples by headspace-gas chromatography, Chromatographia 81 (2018) 1317–1323
Minor comments: Please do not start sentences with numbers (for example Section 2.6). Table 1: present the percentages in decimals (example, 26.42 ± 7.24). The percentage decimal accuracy in a GC-MS determination is to be restricted to one (example, 26.4 ± 7.2).
Necessary corrections have been made throughout the text.
Reviewer 2 Report
Comments and Suggestions for Authors
The authors present a manuscript which looks to link VOCs to inflammation in IBD within a rat model. This is an interesting subject area. I have a few comments
I'm not sure that the abbreviation DSS is explained in the title/abstract
The abstract could have a more quantitative summary of the results i.e. exactly what was found and how did it correlate to disease status. Currently it is written in very general terms.
For example the authors write that the ratios change, but don't give the actual ratios/change etc.
The introduction is quite well written from my perspective at least in dealing with the background to the disease and use of animal models/chemically induced colitis etc. It is perhaps lacking a complete overview of metabolites,VOCs linked to IBD etc. The authors mention changes in ratios of compounds, I feel they could more specifically summarise the previous work in this area.
Methods - should the ethics statement be part of the methods and if it is, should it appear before the description of the animal experiments?
Is the description of a test for occult blood correct if it uses an assessment of colour. I thought occult b lood was a test for hidden blood within stool other matrices?
Can the authors explain a ball-based scoring system - despite the fact it is referenced to another paper probably requires expanion/better description to aid understanding.
This part of the data processing step is not that clearly described in my opinion " The volatile compound percentages were estimated by summing up the percentage of confidently identified compounds for each sample in the AMDIS database. The resulting values were then recalculated as a percentage of the total reliably identified compounds. These conversions were required to avoid content estimation errors due to unreliable matrix signals caused by noise."
This part of the statistical nalaysis also requires further explanation " Prior to PCA, standard procedures were executed. A chi-squared test was used to identify a method to complete missing values. The missing at random (MAR) criteria were identified in the data. The BPCA method was determined to be the most suitable. Subsequently, normalization and scaling procedures were executed."
Results
The DAI needs a more consitent description throughout the paper "The total score of DAI was consisted of body weight decrease, stool consistency and rectal bleeding [22]."
In table 1 why do the authors use a comma to separate the numerical values.
Figure 2A and B show quite an overlap between the control and 7/14 day samples. It would surely be difficult t classify unknown samples in that case.
Figure 2 legend - this is not completely clear "At day 14, the amounts of metabolites identified in the small number of samples are indicated by dashes and were not analyzed for significance."
Needs a better explanation and to number the table referred to "Differences in the metabolomic profile were also recorded on day 7. The resulting table displays the 16 most common compounds meeting these criteria. The main changes affected such compounds as acetic acid, propanoic acid, propanoic acid 2-methyl-, butanoic acid, hexanoic acid (Figures 3 a,b)"
Was this expected? What are the possible reasons? "Therefore, according to the VOC metabolome data, it can be assumed that the microbiota structure continued to change up to the 14th day in contrast to the obvious symptoms of colitis, which were subsiding"
Discussion - it is probably worth the authors discussing the limitation of the model in terms of comparing to humans coupled with a more gradual onset of the disease i.e. if not chemically induced.
Also it would be good to understand better the future directions and implications of the work towards the end of he discussion.
Comments on the Quality of English Language
There are places where the English could be improved, but generally it is fine
Author Response
We are very thankful for the detailed analysis of our research. Our detailed answers are listed below in italic. Please see the attached revised version of our article for corrections indicated by line number.
- I'm not sure that the abbreviation DSS is explained in the title/abstract
Abbreviation DSS is explained in abstract – Line 24
- The abstract could have a more quantitative summary of the results i.e. exactly what was found and how did it correlate to disease status. Currently it is written in very general terms. For example, the authors write that the ratios change, but don't give the actual ratios/change etc.
We provided necessary information in abstract. – Line 31-38
- The introduction is quite well written from my perspective at least in dealing with the background to the disease and use of animal models/chemically induced colitis etc. It is perhaps lacking a complete overview of metabolites, VOCs linked to IBD etc. The authors mention changes in ratios of compounds, I feel they could more specifically summarise the previous work in this area.
We performed more detailed description of available results of VOCs analysis in IBD and also provided the results of microbiota and metabolome assay in a model of DSS-induced colitis in mice. Line – 69-78; Line - 102-113
The following references were added:
- Ahmed I, Greenwood R, Costello B, Ratcliffe N, Probert CS. Investigation of faecal volatile organic metabolites as novel diagnostic biomarkers in inflammatory bowel disease. Aliment Pharmacol Ther. 2016;43(5):596-611. doi:10.1111/apt.13522
- Walton C, Fowler DP, Turner C, et al. Analysis of volatile organic compounds of bacterial origin in chronic gastrointestinal diseases. Inflamm Bowel Dis. 2013;19(10):2069-2078. doi:10.1097/MIB.0b013e31829a91f6
- Wang JL, Han X, Li JX, et al. Differential analysis of intestinal microbiota and metabolites in mice with dextran sulfate sodium-induced colitis. World J Gastroenterol. 2022;28(43):6109-6130. doi:10.3748/wjg.v28.i43.6109
- Methods - should the ethics statement be part of the methods and if it is, should it appear before the description of the animal experiments?
Ethics Statement is the first part of methods section – Line 286-289
- Is the description of a test for occult blood correct if it uses an assessment of colour. I thought occult blood was a test for hidden blood within stool other matrices?
We detailed the description information about DAI – Line - 298-299; Line - 310-315
Disease activity index (DAI) of the rats was estimated by the score of body weight loss, stool consistency and the presence of visible blood in the stool.
- Can the authors explain a ball-based scoring system - despite the fact it is referenced to another paper probably requires expanion/better description to aid understanding?
We evaluated DAI scores according method described previously with several modification. Animal weight loss was calculated as follows: the difference of first day weight and the day of observation was divided by the initial weight multiplied by 100%. References with mentioned calculated method are numbered as 1, 17, 19, 20. We provided additional links [24,25] to available literature data where mentioned DAI calculating method is used - line 126.
- This part of the data processing step is not that clearly described in my opinion " The volatile compound percentages were estimated by summing up the percentage of confidently identified compounds for each sample in the AMDIS database. The resulting values were then recalculated as a percentage of the total reliably identified compounds. These conversions were required to avoid content estimation errors due to unreliable matrix signals caused by noise."
Detailed description of compounds annotation is provided in methods section. Line - 346-356
- This part of the statistical analysis also requires further explanation " Prior to PCA, standard procedures were executed. A chi-squared test was used to identify a method to complete missing values. The missing at random (MAR) criteria were identified in the data. The BPCA method was determined to be the most suitable. Subsequently, normalization and scaling procedures were executed."
Several crucial steps were taken in the data preparation phase before applying Principal Component Analysis (PCA) in our data analysis. Firstly, missing values in the dataset were addressed through a systematic approach. The Chi-square test, a statistical method commonly used to assess relationships between categorical variables, was used to determine an appropriate strategy for dealing with these missing values. Upon conducting the Chi-squared test, it appears that the missing data follows the Missing at Random (MAR) criteria. This is an important factor to consider as it affects the choice of method for imputing missing values. After careful consideration, the Bayesian Principal Component Analysis (BPCA) method was chosen as the most appropriate approach in this context. BPCA is a probabilistic technique that can be used for dimensionality reduction. It is well-suited to address the nature of the missing data identified through the MAR criteria. Its implementation can serve the dual purpose of addressing missing values and preparing the dataset for PCA. Additionally, feature scaling and normalization were executed to enhance the performance of PCA. Normalization ensures that all features are equally considered in the analysis, preventing those with larger scales from having a disproportionate impact on the results.
More detailed description of statistical analysis method was added to the method section - LINE 366-373
Results
- The DAI needs a more consitent description throughout the paper "The total score of DAI was consisted of body weight decrease, stool consistency and rectal bleeding [22]."
Detailed description of DAI was provided in results section - Line 127-133; 148-159
- In table 1 why do the authors use a comma to separate the numerical values.
Necessary corrections have been made in table 1 - Line 174
- Figure 2A and B show quite an overlap between the control and 7/14 day samples. It would surely be difficult t classify unknown samples in that case.
Sample classification based on fecal analysis was not the primary focus of this particular experiment. The model and actual inflammation differ primarily in the duration and depth of the pathophysiological process. The constructed plots aim to demonstrate that primary and short-term intestinal inflammation has an impact on the fecal composition of animals as the PCA is unsupervised and focuses on overall variance. It is important to note that in this model, the gut microbiome has not yet undergone significant changes that would allow short-chain acids to be used as a marker for classifying unknown samples and making a diagnosis (Nguyen, T., Vieira-Silva, S., Liston, A., & Raes, J. (2015). How informative is the mouse for human gut microbiota research?. Disease Models & Mechanisms, 8, 1 - 16. https://doi.org/10.1242/dmm.017400).
- Figure 2 legend - this is not completely clear "At day 14, the amounts of metabolites identified in the small number of samples are indicated by dashes and were not analyzed for significance."
The commentary on figure 2 has been revised to provide a more precise description. Line – 177-180
- Needs a better explanation and to number the table referred to "Differences in the metabolomic profile were also recorded on day 7. The resulting table displays the 16 most common compounds meeting these criteria. The main changes affected such compounds as acetic acid, propanoic acid, propanoic acid 2-methyl-, butanoic acid, hexanoic acid (Figures 3 a,b)"
The typo in the number of metabolites in table has been changed, and a reference to the relevant table has been added to the sentence. Also, the description has been expanded and clarified. Line 183-189
- Was this expected? What are the possible reasons? "Therefore, according to the VOC metabolome data, it can be assumed that the microbiota structure continued to change up to the 14th day in contrast to the obvious symptoms of colitis, which were subsiding"
The mucin layer is the main habitat of adhesive bacteria, including a significant number of Bacteroides and other symbionts. Populations of microorganisms are normally in dynamic equilibrium, so when the bacterial composition alter, the equilibrium shifts towards to more adaptable bacterial species. Histological assay without special dyes does not allow us to assess the degree of mucin layer repair after the regular symptoms of colitis disappearance. In this case, metabolites may indicate an alteration in microbiome composition due to the inferiority of the mucin layer. Therefore, metabolome data allow us to conclude that there are ongoing changes in the structure of the microbiome, which, most likely, should return to its original state after full mucin layer recovery
- Discussion - it is probably worth the authors discussing the limitation of the model in terms of comparing to humans coupled with a more gradual onset of the disease i.e. if not chemically induced.
We provide the limitation of the animal’s model to the discussion section – Line 272-278
- Also it would be good to understand better the future directions and implications of the work towards the end of he discussion.
Future directions and implications are detailed in discussion section – Line 277-284
Round 2
Reviewer 2 Report
Comments and Suggestions for Authors
The authors made changes to the manuscript in line with my comments. From my perspective I am happy that the manuscript could be considered for publication in IJMS.